# Evolutionary Analysis and Functional Identification of Clock-Associated *PSEUDO-RESPONSE REGULATOR* (*PRRs*) Genes in the Flowering Regulation of Roses

**DOI:** 10.3390/ijms23137335

**Published:** 2022-06-30

**Authors:** Abdul Jalal, Jinrui Sun, Yeqing Chen, Chunguo Fan, Jinyi Liu, Changquan Wang

**Affiliations:** College of Horticulture, Nanjing Agricultural University, Nanjing 210095, China; agriculturist.201@gmail.com (A.J.); 15646705902@163.com (J.S.); 2019204053@njau.edu.cn (Y.C.); 2018104110@njau.edu.cn (C.F.); jyl@njau.edu.cn (J.L.)

**Keywords:** clock *PRRs*, evolution, flowering, rose

## Abstract

Pseudo-response regulators (*PRRs*) are the important genes for flowering in roses. In this work, clock *PRRs* were genome-wide identified using Arabidopsis protein sequences as queries, and their evolutionary analyses were deliberated intensively in Rosaceae in correspondence with angiosperms species. To draw a comparative network and flow of clock *PRRs* in roses, a co-expression network of flowering pathway genes was drawn using a string database, and their functional analysis was studied by silencing using VIGS and protein-to-protein interaction. We revealed that the clock *PRRs* were significantly expanded in Rosaceae and were divided into three major clades, i.e., *PRR5/9* (clade 1), *PRR3/7* (clade 2), and *TOC1/PRR1* (clade 3), based on their phylogeny. Within the clades, five clock *PRRs* were identified in *Rosa chinensis*. Clock *PRRs* had conserved RR domain and shared similar features, suggesting the duplication occurred during evolution. Divergence analysis indicated the role of duplication events in the expansion of clock *PRRs*. The diverse cis elements and interaction of clock *PRRs* with *miRNAs* suggested their role in plant development. Co-expression network analysis showed that the clock *PRRs* from *Rosa chinensis* had a strong association with flowering controlling genes. Further silencing of *RcPRR1b* and *RcPRR5* in *Rosa chinensis* using VIGS led to earlier flowering, confirming them as negative flowering regulators. The protein-to-protein interactions between *RcPRR1a*/*RcPRR5* and *RcCO* suggested that *RcPRR1a*/*RcPRR5* may suppress flowering by interfering with the binding of *RcCO* to the promoter of *RcFT*. Collectively, these results provided an understanding of the evolutionary profiles as well as the functional role of clock *PRRs* in controlling flowering in roses.

## 1. Introduction

The circadian clock is a time-keeping mechanism in a wide range of organisms controlling endogenous biological rhythms to adapt to 24 h day-night cycles [1,2]. The circadian clock of land plants consists of multiple interconnected transcriptional feedback loops [3,4], wherein the sequential expression of core circadian components assist plants in predicting daily changes in zeitgebers via rhythmic expression of circadian target genes [2]. The first molecular model of circadian clock proposed in *Arabidopsis thaliana* is comprised of a negative transcriptional–translational feedback loop including two MYB-like transcription factors *CIRCADIAN CLOCK-ASSOCIATED (CCA1)* and *LATE ELONGATED HYPOCOTYL (LHY)*, and their homologs, *REVEILLE8 (RVE8/LHY-CCA1-LIKE5/LCL5)* and *RVE4*, as well as *PRR1, PRR3, PRR5, PRR7,* and *PRR9* [5,6,7,8,9,10]. In the circadian rhythm of Arabidopsis, the translation of *CCA1* and *LHY* occurs in the morning time and mutually interacts to repress the expression of *TOC1*/*PRR1* [5,11,12]. The sequential expression of clock *PRR5*, *PRR7*, and *PRR9* during daytime form an additional loop showing their partial redundant role in repressing the transcription of *CCA1* and *LHY* and are considered the homologs of *TOC1*/*PRR1* [9]. *PRR* genes are very substantial in plant circadian rhythm in terms of flowering time, development of inflorescence architecture, and transition of vegetative to reproductive phase [13]. *PRR* genes are highly conserved in the circadian rhythm of Arabidopsis (*TOC1*/*PRR1*, *PRR3*, *PRR5*, *PRR7*, and *PRR9*) and rice (*OsPRR1, OsPRR37, OsPRR73, OsPRR95,* and *OsPRR59*), and both species have the same number of *PRR* gene family members with RR domain at N-terminal and CCT domain on C-terminal [8,14,15]. *PRR* genes have received attention in both dicots such as Arabidopsis [16,17], soybean [18], and monocots such as rice [19], wheat [20,21,22], barley [23], sorghum [24], maize [25], yet their evolutionary profiles and biological functions especially related to flowering regulation in woody plants remain largely unknown.

Roses are globally important ornamental plants having a complex and long history of domestication. Roses are commonly found in growing gardens or in vases as cut flowers. Roses are the well-known treasured flowers in the history of mankind, and it has been grown since before the Common Era. Roses that are originally grown for medicinal purposes and perfumes eventually become a valued ornamental flowering plants [26]. There are about 150 wild-growing species of roses in the northern hemisphere alone and may be tens of thousands of rose cultivars grown to date. However, only 8 to 20 of these have been involved in the breeding of four main currently cultivated lineages [27]. Nowadays, rose flower cultivars are becoming more diverse, ranging from more traditional varieties with tall, pointed petals to those with rounded, cup-shaped petals; quarter-shaped petals similar to those of the old rose; or single-petal flowers [26]. Flowering remains the key life event of roses and is greatly affected by various endogenous and environmental signals. Due to continuous flowering behavior, roses have attained special attention with respect to their commercial value and use in the landscape. The release of the genome sequences of 24 Rosaceae species covering 2 major lineages (https://www.rosaceae.org/, accessed on 30 April 2022) [28] enabled us to comprehensively study the evolutionary and functional profiles of circadian clock-associated *PRR* genes (clock *PRRs*) in roses.

In this study, we performed genome-wide identification of clock *PRRs* in Rosaceae in relation to basal angiosperms species, covering the major subfamilies (Rosoideae and Amygdaloideae) of the order Rosales, and initiated a systematic phylogenetic analysis to obtain the overall evolutionary picture of clock *PRRs* in Rosaceae. We further investigated conserved protein motifs, domains, and gene structure organization by comparative analysis of the protein sequences of clock *PRRs* to understand their function in roses. Moreover, we also performed co-expression network construction and virus-induced gene silencing (VIGS) of clock *PRRs* in *Rosa chinensis*. Our study provided an understanding of the evolutionary profiles as well as the functional role of clock *PRRs* in controlling flowering in roses.

## 2. Results

### 2.1. Clock PRRs Expanded in Rosaceae (Perennial Species) as Compared with Brassicaceae (Annual Species) and Basal Angiosperm

To gain insight into the identification and evolution of clock *PRRs* in Rosaceae in comparison with basal angiosperms, the Arabidopsis clock *PRRs* protein sequences were used as queries (See Materials and Methods (Section 4)) against the whole genomes of all the representative species (Figure 1A). We identified 10 clock *PRRs* in 2 species of basal angiosperms, 50 in 8 Brassicaceae species, and 157 clock *PRRs* in 18 Rosaceae species. Within the Rosaceae family, we identified 105 clock *PRRs* in 11 species of subfamily Amygdaloideae and 52 in 7 species of subfamily Rosoideae, respectively. Among them, the copy number of clock *PRRs* ranged from 4 to 7 with a mean value of 6.3 in Brassicaceae species, while it ranged from 5 to 19 with a significantly higher mean value of 8.7 in Rosaceae species (Figure 1B). Furthermore, the range of copy numbers of clock *PRRs* in subfamilies Amygdaloideae and Rosoideae of Rosaceae were the same (5 to 19), but the mean value for copy number in Amygdaloideae (9.5) was higher than that of Rosoideae (7.4) (Figure 1C). Moreover, clock *PRRs* in two representative species of basal angiosperms were 3 in *Amborella trichopoda* and 7 in *Nymphaea colorata*. These results indicated that in comparison with basal angiosperms and Brassicaceae, the clock *PRRs* were significantly expanded in Rosaceae during evolution.

### 2.2. All Rosaceae Clock PRRs Were Classified into Three Clades

To obtain a deeper insight into the expansion profiles of clock *PRRs* in Rosaceae, a maximum likelihood tree (see Materials and Methods (Section 4)) was constructed from a total of 239 identified clock *PRRs* proteins. Based on the topology of the phylogenetic tree, the evolutionary analysis revealed that the clock *PRRs* of Rosaceae had three major clades, including *PRR5/9* (clade1), *PRR3/7* (clade2), and *TOC1/PRR1* (clade3) (Figure 2 and Appendix A). Clade 1 was orthologous of *AtPRR5* and *AtPRR9*, clade 2 was orthologous of *AtPRR3* and *AtPRR7*, and clade 3 was orthologous of *AtPRR1* (*TOC1*) of *Arabidopsis thaliana*. The evolutionary insight further verified that, similar to Brassicaceae, clade 1 was the major clade, followed by clade 2 and clade 3 in Rosaceae. Furthermore, in subfamily Rosoideae, clade 1 was the major clade, followed by clade 3 and clade 2, just like basal Rosids, while in subfamily Amygdaloideae, the major clade was clade 2, followed by clade 1 and clade 2 such as basal angiosperms (Figure 2 and Appendix A). On the basis of these comparative phylogenetic relationships, it was revealed that the clock *PRRs* in Rosaceae were clustered in the three clades as that of *Arabidopsis thaliana*, supporting the previously described lineage.

### 2.3. Response Regulator (RR) Domains Were Conserved in Clock PRRs of Roses during Evolution

On the basis of the comparative phylogenetic relationship, another phylogenetic tree for *Rosa chinensis* with neighbor-joining model was constructed using the clock *PRR* of *Arabidopsis thaliana* and *Rosa chinensis* via MEGA 11 software (Appendix A). The basic gene parameters of identified clock *PRRs* of *Rosa chinensis* were exhibited in Appendix A, including gene ID, location on the chromosome, gene length (AA), molecular weight (Mw), isoelectric point (pI), and predicted subcellular localization. To further support the phylogenetic reconstruction, the identified five clock *PRRs* from *Rosa chinensis*, named *RcPRR1a*, *RcPRR1b*, *RcPRR5*, *RcPRR3*, and *RcPRR7* (Appendix A), were further investigated for conserved domains and motifs, as well as for gene structure organization and chromosomal localization. All the clock *PRRs* had both RR (response regulator) domain and REC domain at N-terminal and CCT domain at C-terminal, except for *RcPRR1b,* which had no CCT domain (Figure 3B). Similarly, five conserved motifs corresponding to RR and REC domains were present in all clock *PRRs,* whereas motif 2 corresponding to the CCT domain was missing in *RcPRR1b* (Figure 3A). Logos of the identified 5 motifs of all clock *PRRs* were shown in Figure 3D. Furthermore, gene structure organizations of the clock *PRRs* were further illustrated (Figure 3C). The figure demonstrated the distribution of coding regions (CDS) and untranslated regions (UTRs) of clock *PRRs*, indicating that the exons and introns were highly diverse in all genes. *RcPRR5*, *RcPRR3*, and *RcPRR7* had 8 exons, and *RcPRR1a* had 6 exons; however, *RcPRR1b* appeared to be the shortest one among all *PRRs* with 4 exons. Moreover, the UTRs were also detected in all clock *PRRs* of *Rosa chinensis*. The chromosomal localization further revealed that *RcPRR3* and *RcPRR5* were present on chromosome 1, *RcPRR7* was present on chromosome 3, while *RcPRR1a* and *RcPRR1b* were present on chromosome 7 of *Rosa chinensis* (Figure 3E). These results suggested that RR domains were conserved in roses and thus shared some similar features.

### 2.4. Determination of Non-Synonymous (Ka) and Synonymous (Ks) Substitution Rate and Cis-Regulatory Elements Analysis of Clock PRRs in Roses

To find the evolutionary aspects of clock *PRRs*, a divergence analysis was performed. The non-synonymous substitution per non-synonymous site (*Ka*) and synonymous substitution per synonymous site (*Ks*) were determined for two paralogous clock *PRRs* based on the phylogenetic tree generated via *Ka/Ks* calculation server to see the evolutionary discretion among clock *PRRs* (Table 1). The *Ka/Ks* value (<1) of the two pairs of genes indicated the purifying selection pressure during the evolution. The divergence time (T) for both pairs of genes ranged from 25.1 to 42.5 million years ago (MYA).

The promoter region analysis of rose clock *PRRs* for the presence of *cis*-regulatory elements resulted in a diverse range of *cis* elements. The *cis* elements were classified into four major categories, including light-responsive elements, hormone-responsive elements, stress-responsive elements, and the elements involved in plant growth and development. The results revealed that light-responsive elements were G-box, GT1-motif, Sp1, and 3-AF1 binding site, and hormone-responsive elements were TGACG-motif, TGA-element, TCA-element, P-box, AuxRR-core, ABRE, GARE-motif, CGTCA-motif, and TATC-box, stress-responsive elements were LTR, MBS, and TC-rich repeats, the responsive elements for plant growth and development were ARE, GCN4_motif, and CAT-box (Figure 4).

### 2.5. Interaction of PRRs with Chemical Compounds and Known miRNAs

To identify the interacting *miRNAs* targeting the clock *PRRs*, the coding sequences clock *PRRs* were used against the *miRNAs* of *Rosa chinensis*. It is revealed that 27 *miRNAs* showed interaction with all clock *PRRs* except *RcPRR3* (Figure 5). The results further revealed that 17 *miRNAs* alone showed interaction with *RcPRR5*, four with *RcPRR7*, 3 with *RcPRR1a,* and another 3 with *RcPRR1b*. Although all the five clock *PRRs* have a strong interaction among them, however, the *miRNAs* only targeted four clock *PRRs,* namely, *RcPRR1a*, *RcPRR1b*, *RcPRR5*, and *RcPRR7*. The Excel spreadsheet containing *miRNAs* ID, targeting sites, and alignment with clock *PRRs* are given in the Appendix A.

To further identify the interaction of some chemical compounds with clock *PRRs*, the network of clock *PRRs* with chemical compounds was generated via Cytoscape (Figure 6). The five clock *PRRs* showed interaction with 5 different phyto-elements. *RcPRR1a*, *RcPRR3*, and *RcPRR7* collectively interacted with auxin. *RcPRR1b* and *RcPRR7* showed interaction with ascisic acid, salicylic acid, gibberellin, and methyl jasmonate (MeJA). *RcPRR3* showed interaction with ascisic acid, salicylic acid, and gibberellin, along with auxin. *RcPRR5* showed interaction with salicylic acid, gibberellin, and MeJA but strong interaction with gibberellin.

### 2.6. Co-Expression Network of Clock PRRs and Flowering Pathway Genes in Rosa Chinensis

To obtain insight into the association of clock *PRRs* with flowering control genes in roses, the identified clock *PRRs* were further analyzed via string software (see Materials and Methods (Section 4)) to obtain a visualized map (Figure 7). The transcripts of *Rosa chinensis* were recognized in the string database with known annotation based on possible co-expressions. It further confirmed that the association of clock *PRRs* with flowering control genes was not random and connected with a highly significant value of *p* < 1 × 10^−16^. The co-expression networks were further analyzed for protein-protein interactions (edges/lines) and the shared biological processes. The addition of edges for connecting nodes showed a more significant interaction, while the reducing number of edges represented less interaction among proteins. It was observed that all clock *PRRs* had a strong association with each other and with another clock-associated gene, *CCA1*.

Moreover, all clock *PRRs* genes shared some identical biological processes with flowering pathways genes. The results revealed that all the 13 genes (5 clock *PRRs* and 8 flowering pathway genes) together mainly contributed to the regulation of biological processes (e-value 1 × 10^−6^). Within these 13 genes, all the 5 clock *PRRs*, *CCA1*, *CO*, *FT1*, and *FT2,* were closely connected in controlling flowering based on the KEGG pathways association of circadian rhythm (2.7 × 10^−20^). *CCA1* had a strong interaction with clock *PRRs* genes and was involved in photoperiodic flowering (e-value 9 × 10^−5^) and rhythmic process (e-value 3 × 10^−9^). *FT-1* and *FT-2* were involved in inflorescence development, and the regulation of timing of the transition from vegetative to reproductive phase (e-value 0.00), *SOC1*, *FT1*, *FT2*, and *CO* were mutually involved in the regulation of flower development (e-value 5 × 10^−5^). *RcPRR1a*, *RcPRR1b, SOC1, FUL*, *FT1*, and *FT2* together affected the flower development (e-value 1 × 10^−5^) and reproductive structure development (2 × 10^−5^). To obtain a deeper understanding of the co-expression network, the KEGG pathway of the circadian rhythm was visualized via a direct link of KEGG genome pathways from the String database (https://www.genome.jp/pathway/rcn04712) (accessed on 29 March 2022) (Appendix A). Taken together, these results suggested that clock *PRRs* of *Rosa chinensis* work together with flowering controlling genes.

### 2.7. Silencing of RcPRR1a and RcPRR5 Promoted Flowering in Rosa Chinensis

To further identify the functional role of clock *PRRs* in controlling flowering, we silenced *RcPRR1a* and *RcPRR5* in *Rosa chinensis,* followed by checking the flowering phenotype and the expression levels of flowering regulating genes (*RcCO* and *RcFT*). qRT-PCR analysis revealed that the expression levels of *RcPRR1a* and *RcPRR5* in silenced lines were significantly reduced (Figure 8C), and both *RcPRR1a* and *RcPRR5* silenced lines flowered earlier as compared with the control plants (Figure 8A,B). Consistently, the expression of flower regulating gene *RcFT* (Figure 8E) was upregulated as compared with the control. These results directed that clock *PRRs* had influenced the floral integrator and suppressed flowering. Moreover, no significant increase in the expression level of *RcCO* was found in silenced lines (Figure 8D). This alerted us that *RcPRR1a* and *RcPRR5* might have protein interactions with *RcCO* to interfere with its function during flowering in *Rosa chinensis*.

### 2.8. RcPRR1a/RcPRR5 Physically Interacted with RcCO

To further determine the possibility of *RcPRR1a* and *RcPRR5* genes interfering with *RcCO* during flowering, we examined the protein-to-protein interactions between *RcPRR1a/RcPRR5* and *RcCO*. We performed split luciferase (LUC) complementation assays by fusing *RcPRR1a* and *RcPRR5* separately to the N-terminal and *RcCO* to the C-terminal fragments of luciferase, respectively. We infiltrated *Agrobacterium tumefaciens* cells with these constructs into *Nicotiana benthamiana* leaves. Luciferase activity was detected only in *N. benthamiana* leaves co-infiltrated with both *35S:RcPRR1a-N:LUC* and *35S:RcCO-C:LUC* or *35S:RcPRR5-N:LUC* and *35S:RcCO-C:LUC* but not in leaves infiltrated with the N- or C-terminal fragment of *RcPRR1a*, *RcPRR5* or *RcCO* alone (Figure 9). These results confirmed the interaction of *RcPRR1a*/*RcPRR5* with *RcCO* and suggested the *RcPRR1a*/*RcPRR5* may decrease the free pool of *RcCO* and interfere with its binding to the *RcFT* promoter during flowering.

## 3. Discussion

The circadian clock of plants is referred to an endogenous oscillator regulating various plant physiological processes such as photomorphogenesis, stress responses, and flowering [28,29,30,31,32,33,34,35,36,37,38]. *PRR* genes are very important in plant flowering in response to circadian rhythm [13]. In many plants, the clock *PRR* genes are highly conserved. In *A. thaliana*, *Aegilops Tauschii*, *Hordeum vulgare*, *Sorghum bicolour*, *Triticum aestivum*, and *Oryza sativa*, there are five members of clock *PRR* genes [23,39,40,41,42,43]. However, the evolutionary history of *PRR* genes and their functional identification in Rosaceae remain unknown.

With the release of whole-genome sequences of Rosaceae, we genome-wide identified the *PRR* genes in Rosaceae in correspondence with angiosperms species and demonstrated their evolutionary features along with their functional identification. The clock *PRRs* were significantly expanded in Rosaceae during evolution (Figure 1) but had conserved RR domains, indicating their potential functional conservation in angiosperms as previously described in monocots and dicots species [13,16,42]. The phylogenetic analysis showed that the clock PRRs could be classified into three major clades, including *PRR5*/*9* (clade1), *PRR3/7* (clade2), and *TOC1/PRR1* (clade3) (Figure 2), and the clock *PRRs* within each clade were highly correlated to each other [13,42,44,45,46]. These results facilitated the understanding of *PRRs* regarding their functional conservation and evolution in Rosaceae.

To explore the evolutionary correlations of rose clock *PRRs*, the phylogenetic reconstruction of roses revealed the three major clads (Appendix A) that have also been reported in various earlier studies [44,46]. The further results from domains, conserved motifs, and gene structure further confirmed the structural conservation of clock *PRRs* in rose plants (Figure 3), while the divergence analysis (Table 1) revealed purifying selection pressure of clock *PRRs* during the evolution [47]. The *PRR* family proteins appeared to be unique to plants [48], having an RR domain at N-terminal [8] followed by the additional C-terminal (CCT) motif that could also be found in the *CO (CONSTANS)* transcription factor [49]. The structure similarity implied the function resemblance, which is further supported by *cis*-elements analysis (Figure 4), interaction with *miRNAs* (Figure 5) [50,51], and the co-expression network and functional annotation of clock *PRRs* with flower regulating genes (Figure 7).

Moreover, virus-induced gene silencing (VIGS) of *RcPRR1a* and *RcPRR5* induced early flowering in *Rosa chinensis*, providing the genetic evidence for the function of clock *PRRs* in rose flowering regulation (Figure 8). Our results were inconsistent with the previously described role of clock *PRRs* as negative flowering regulators in rice [52], sorghum [41], and soybean [18]. In Arabidopsis, the decreasing of *At**PRR**5* was diligently associated with late flowering [52,53,54], defining it as flowering activators. However, in rice, the overexpression of the *At**PRR**5* homolog *OsPRR58* [52] delayed flowering, while silencing of the *At**PRR**3* homolog *OsPRR37* promoted flowering [55]. The decreasing of *GmPRR37* in soybean [18] also accelerated the flowering, in line with the result of *OsPRR37* in rice. Similarly, the nonfunctional allele of *SbPRR37* (*At**PRR**3* homolog) attenuated the expression of *CO* but upregulated the flowering activators *Ehd1*, *FT*, and *ZCN8* [41], suggesting *SbPRR37* as a flowering repressor in sorghum. These results indicated the functional divergence of clock *PRRs* in long-day and short-day plants [18]. In the present study, there was a negative correlation between *PRR1a**/**RcPRR5* and *FT* [55] (Figure 8). We hypothesized that clock *RcPRRs* may form complexes with *RcCO* to control *FT* transcription [56]. The further results of protein-protein interaction analysis confirmed the protein interaction of clock *PRRs* with *CO* (Figure 9), showing that *RcPRR1a*/*RcPRR5* may decrease the free pool of *RcCO* and interfere its binding to *RcFT* promoter during flowering.

## 4. Materials and Methods

### 4.1. Data Source and Sequence Retrieval

The genome data of most Rosaceae species were retrieved from GDR (https://www.rosaceae.org/). In addition, the genome data of *Fragaria* species were downloaded from Strawberry GARDEN (http://strawberry-garden.kazusa.or.jp/), and the genome of *Prunus mume* was downloaded from NCBI/Genome (https://www.ncbi.nlm.nih.gov/genome). The *Rosa multiflora* genome was downloaded from a specific genome portal (http://rosa.kazusa.or.jp/index.html), and different versions of the *Rosa chinensis* genome were downloaded from the specific genome portals (https:// iris.angers.inra.fr/obh/ for V1 and https://lipm-browsers.toulouse.inra.fr/pub/RchiOBHm-V2 for V2). The latest version of genome data of Brassicaceae species and other species of basal angiosperms were downloaded from Phytozome v12 (https://phytozome.jgi.doe.gov/pz/portal.html), and *Vitis vinifera* were downloaded from CRIBI (http://genomes.cribi.unipd.it/) (All accessed on 1 September 2021).

### 4.2. Clock PRRs Orthologous Identification in Roses and Angiosperms

The Arabidopsis clock-associated *PRRs* protein sequences, i.e., *PRR1* (*At5G61380*), *PRR3* (*At5G60100*), *PRR5* (*At5G24470*), *PRR7* (*At5G02810*), and *PRR9* (*At2G46790*), were used as queries in BLASTP searches against the protein sequences of all the representative species of basal angiosperms, basal rosids (Vitals), rosaceae (Fabids), and brasicaceae (Malvids) families. The evolutionary analysis was carried out for basal angiosperms vs. Rosids. Within the Rosids, the species were selected from each subgroup to obtain profound results for Rosaceae. All the sequences with an e-value threshold of 1 × 10^−3^ were extracted as candidate clock *PRRs* orthologs. For further assurance of the clock *PRRs* orthologs, another BLASTP search was performed using the candidate orthologs as queries against the whole genome of Arabidopsis set as a database. The candidate orthologs with Arabidopsis clock *PRRs*, as best hits, were identified as clock *PRRs* orthologs in all the representative plant species of basal angiosperms and Rosids. As Arabidopsis clock *PRRs* belong to the response regulator (RR) gene family and are among the 32 members RR family (A-type response regulator gene family having 11 members, B-type response regulator gene family having 12 members, and pseudo-response regulator gene family having 9 members) (https://arabidopsis.org/browse/genefamily/ARR.jsp) (accessed on 7 November 2021), the sequences of all the identified clock *PRRs* orthologs in different species were aligned using MAFFT in corresponding to Arabidopsis RR gene family members and maximum likelihood method implemented in IQ-Tree was used to construct a phylogenetic tree with 1000 bootstraps to identify clock *PRRs* clade. All the sequences identified as clock *PRRs* in all species were further confirmed by performing domain annotation using the Pfam database (http://pfam.janelia.org/), SMART database (http://smart.embl-heidelberg.de/), NCBI conserved domain database (http://www.ncbi.nlm.nih.gov/Structure/cdd/wrpsb.cgi), and MOTIF search (https://www.genome.jp/tools/motif/) (accessed on 20 November 2021) for CCT and RR (Response Regulator) domains. The sequences without RR (Response Regulator) domain or with partial domain sequences (Appendix A) were not used for further analysis [57].

### 4.3. Sequence Alignment and Phylogenetic Analysis

All the sequences with satisfying requirements from basal angiosperms, rosids, rosaceae, and brasicaceae families were aligned for multiple sequence alignments using alignment software MAFFT (v7.037b, Osaka University, Osaka, Japan) [58] with the most accurate alignment strategy of L-INS-I. Maximum likelihood trees were constructed using both the FastTree software with the JTT+CAT model (http://www.microbesonline.org/fasttree/) [59] and IQTREE with the JTT+R5 or JTT+R8 model (http://www.iqtree.org/). The phylogenetic trees were further visualized and edited using MEGA7 software (https://www.megasoftware.net/home) [60] (accessed on 7 November 2021).

### 4.4. Conserved Domains and Motifs Analysis and Gene Structure Organization of Clock PRRs in Roses

For domain analysis, the clock *PRRs* protein sequences of *Rosa chinensis* were subjected to NCBI CDD online software (https://www.ncbi.nlm.nih.gov/Structure/cdd/wrpsb.cgi) (accessed on 6 February 2022), and the predicted information was then used to visualize domain information via TBtools software (https://github.com/CJ-Chen/TBtools) (accessed on 6 February 2022). Similarly, conserved motif analysis of clock *PRRs* proteins was accomplished by MEME diagrams. The protein sequences of clock *PRRs* were submitted to MEME suite software 5.4.1 (https://meme-suite.org/meme/tools/meme) (accessed on 6 February 2022) for 5 conserved motifs identification and then visualized by TBtools software. Correspondingly, the gene structure organization of the clock *PRRs* were also visualized in TBtools software by submitting the gff3 files of *Rosa chinensis* along with the identified gene IDs.

### 4.5. Divergence and Cis-Elements Analysis

The server *Ka/Ks* calculation tool (http://services.cbu.uib.no/tools/kaks) (accessed on 26 June 2022) was used to determine the non-synonymous substitution per non-synonymous site (*Ka*) and synonymous substitution per synonymous site (*Ks*) by inputting the protein DNA sequences of clock PRRs via using default parameters. The divergence time was calculated by the given formula [61]:$$Time\;of\;divergence\;\left(T\right)=\frac{Synonymous\;ubstitution\;rate\;\left(dS\;\mathrm{or}\;Ks\right)}{2\times Divergence\;rate\;(6.56\times 10^{-9})}\times TMY\;(10^{-6})$$

*Cis*-element analysis was performed by selecting an upstream region of 1500 bp of each genomic sequence of the clock PRR gene and was searched for the presence of *cis*-regulatory elements in the PlantCARE server (https://bioinformatics.psb.ugent.be/webtools/plantcare/html/) (accessed on 26 June 2022) [47,62].

### 4.6. Interaction Analysis of PRRs with Chemical Compounds and Known miRNAs

The STRING network was generated for clock *PRRs* using the STRING server (http://stringdb.org) (accessed on 26 June 2022) and was further analyzed by the Cytoscape extension of STRING. The targeting *miRNAs* for clock *PRRs* were recognized by searching the CDS of clock *PRRs* against the published miRNAs of *Rosa chinensis*, downloaded from the specific genome portal (https://lipm browsers.toulouse.inra.fr/pub/RchiOBHm-V2) (accessed on 26 June 2022), through psRNATarget database (accessed on 26 June 2022) and finally visualized via the Cytoscape software (Cytoscape Consortium, USA) (http://apps.cytoscape.org/apps/stringapp) (accessed on 26 June 2022) [51].

### 4.7. Co-Expression Network Organization of Clock PRRs and Flowering Pathway Genes in Roses

The co-expression network of the *PRRs* genes with flowering control pathways genes was generated by using the protein sequences of the clock *PRRs* genes along with the flowering pathways genes via String (String consortium, 2022, version 11.5) (https://cn.string-db.org/) (accessed on 29 March 2022). The protein sequences of important genes involved in the flowering pathway of Arabidopsis were used as queries to BLAST against protein sequences of the *Rosa chinensis* genome, and the sequences with the e-value of 1 × 10^−10^, detected as best hits, were identified as the homologous sequences. These sequences were further used along with *PRRs* for their co-expression analysis and functional annotation in the String database against the available transcriptomic data of *Rosa chinensis,* following the methods previously described by [63].

### 4.8. Virus-Induced Gene Silencing of RcPRR1a and RcPRR5 in Rosa chinensis

To investigate the role of clock *PRRs* in controlling flowering in roses, *RcPRR1a* and *RcPRR5* from *Rosa chinensis* were silenced according to the previously described method [64]. Specific gene fragments of the full-length CDS, selected from *RcPRR1a* and *RcPRR5*, were amplified using the primers given in Appendix A. Vectors were constructed for *pTRV2:RcPRR1a* and *pTRV2:RcPRR5*, while empty pTRV2 was used as a mock. The mixture of *A. tumefaciens* cultures carrying pTRV1 and *TRV2:RcPRR1a*, *pTRV2:RcPRR5* with the ratio of 1:1 (*v*/*v*) or with pTRV1 and pTRV2 (empty used as the mock) was vacuum infiltrated into grown cuttings (stem having two nodes). The infiltrated segments of the stem were carefully washed with distilled water and planted into vermiculite for rooting and branching. Leaf samples were collected in liquid nitrogen after 6–8 weeks of infiltration, and the expression level of silenced genes and flowering-related genes were examined via qRT-PCR using the primers given Appendix A. *RcGADPH* was used as a reference gene as described previously [65]. The phenotypic evaluation was also recorded.

### 4.9. Protein-Protein Interactions Analysis of RcPRR1a and RcPRR5 with RcCO

To assay protein-protein interactions, the CDS of *RcPRR1a* and *RcPRR5* without the stop codons was cloned into pMK7-nL-WG2 (http://www.psb.ugent.be/) (accessed on 13 January 2022), while the CDSs of *RcCO* without the stop codons was cloned into pMK7-cL-WG2. *A. tumefaciens* strain GV3101 cells carrying *35S:RcPRR1a:LUC-N*, *35S:RcPRR5:LUC-N*, and *35S:RcCO:LUC-C* constructs were co-infiltrated into *N. benthamiana* leaf epidermal cells to examine the reconstitution of *LUC*. Three biological replicates were performed for each experiment. Luciferase imaging was performed using a CCD camera (Andor Technology, Belfast, UK). At 48 h after agroinfiltration of *N. benthamiana* leaf epidermal cells, images were acquired every 10 min for 30 min, and luciferase activity was quantified as the mean counts per pixel per exposure time using Andor Solis image-analysis software (Andor Technology, Belfast, UK) [66,67].

## 5. Conclusions

The evolutionary analysis indicated that the clock *PRRs* were significantly expanded in Rosaceae and classified into three major clades *PRR5/9* (clade1), PRR3/7 (clade2), and *TOC1/PRR1* (clade3). It indicated that clock *PRRs* were conserved in Rosaceae and reflected their functional conservation for flowering. Moreover, *Rosa chinensis* was detected to have 5 clock *PRRs* (*RcPRR1a*, *RcPRR1b*, *RcPRR5*, *RcPRR3*, and *RcPRR7*). The domain and motif analysis further confirmed that clock *PRRs* had conserved RR domain, and the similar features of gene structure may be due to the duplication events during evolution. Divergence analysis indicated the role of duplication events in the expansion of clock *PRRs*. To anticipate the functional analysis, *cis*-element analysis, interaction analysis with *miRNAs,* and chemical compounds were performed. Co-expression network of clock *PRRs* showed interaction with flowering regulating genes. Moreover, the phenotypic and genetic evidence of the silenced lines of clock *PRRs* further confirmed the role of clock *PRRs* in flowering regulation in roses. Consistently, the protein interaction of *RcPRR1a* and *RcPRR5* with *RcCO* further explored the involvement of *RcPRR1a* and *RcPRR5* in interfering with *RcCO* binding to the promoter of *RcFT* during flowering. It could be concluded that clock *PRRs* play a crucial role as a critical factor for photoperiodic flowering time.

## Figures and Tables

**Figure 1 ijms-23-07335-f001:**
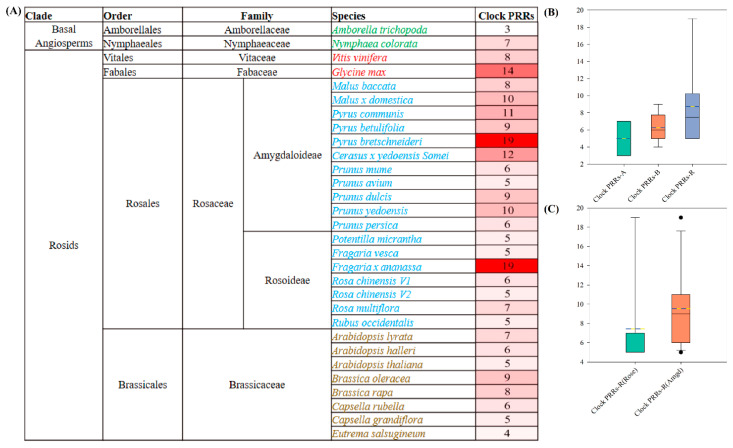
Clock *PRRs* in basal angiosperms, Brassicaceae and Rosaceae. (**A**) List of the copy number of identified clock *PRRs* in our study; (**B**) boxplots of the copy numbers of clock *PRRs* in basal angiosperms, Brassicaceae and Rosaceae. Clock *PRRs*-A, Clock *PRRs*-B, and Clock *PRRs*-R denote the number of clock *PRRs* in Basal Angiosperms, Brassicaceae, and Rosaceae, respectively; (**C**) boxplots of the copy numbers of clock *PRRs* in subfamilies of Rosaceae. Clock *PRRs*-R (Rose) and clock *PRRs*-R (Amgd) denote subfamily Rosoideae and Amygdaloideae, respectively.

**Figure 2 ijms-23-07335-f002:**
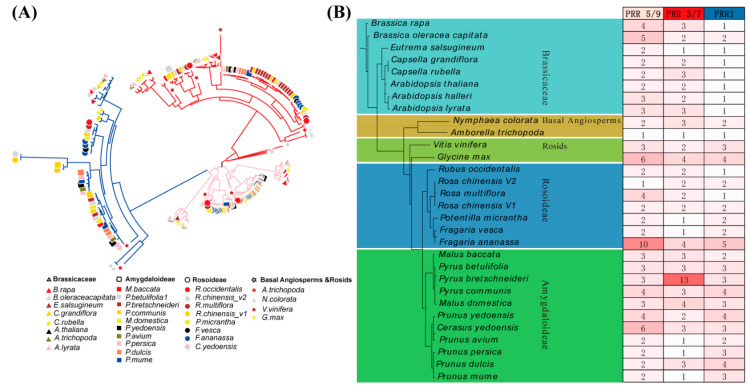
Phylogeny of clock *PRR* genes in Rosaceae. (**A**) Phylogenetic tree of all clock *PRRs* from 2 basal angiosperm species, 18 Rosaceae species, 8 Brassicaceae species, and 2 basal Rosids species. Clock *PRRs* of Rosaceae classified into three major clades on the basis of their phylogeny shown in different branch colors; (**B**) species tree constructed on the basis of clock *PRRs* phylogenetic relation and the number of clock *PRRs* in each clade of every species.

**Figure 3 ijms-23-07335-f003:**
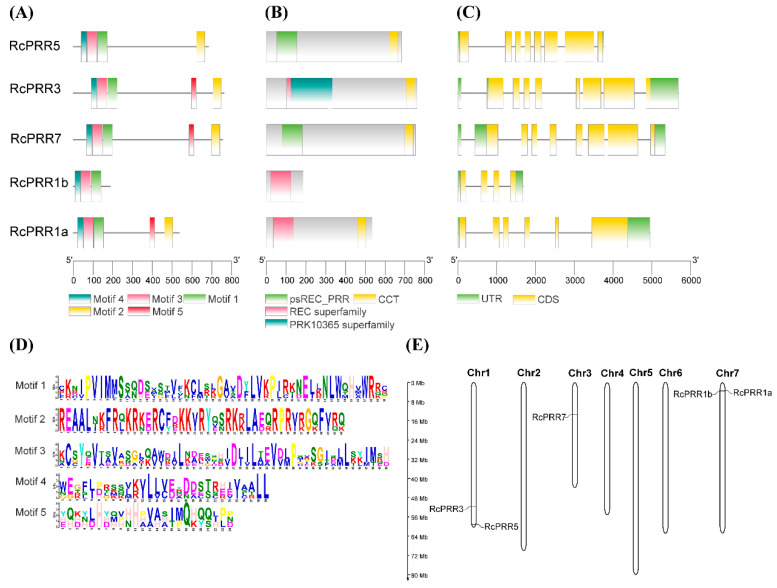
Conserved motifs and domains, gene structure organization, and chromosomal localization of clock *PRRs* in *Rosa chinensis*. (**A**) discovered motifs; (**B**) discovered domains; (**C**) gene structure organization; (**D**) logos of the identified motifs; and (**E**) chromosomal localization.

**Figure 4 ijms-23-07335-f004:**
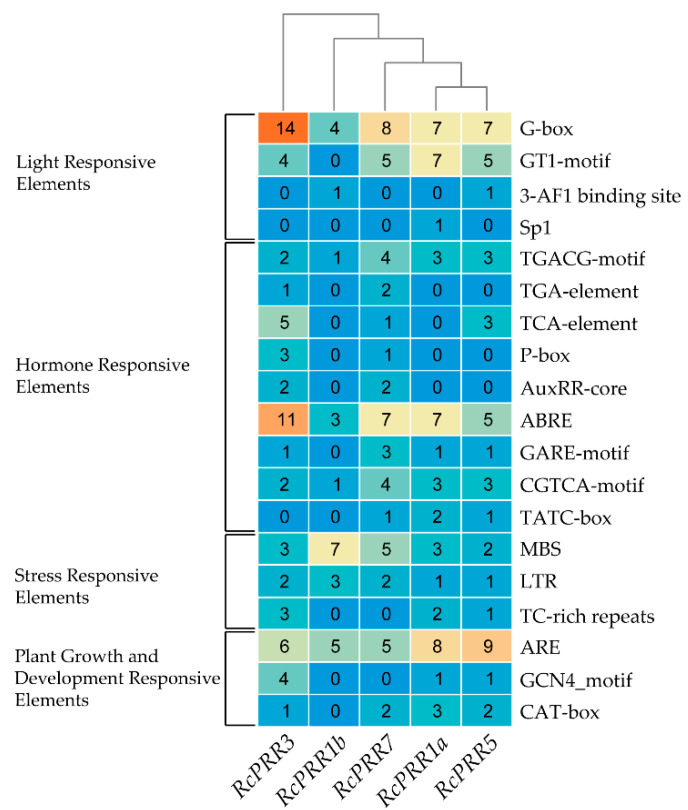
*cis*-regulatory elements present in the promoter region of clock *PRRs* of rose.

**Figure 5 ijms-23-07335-f005:**
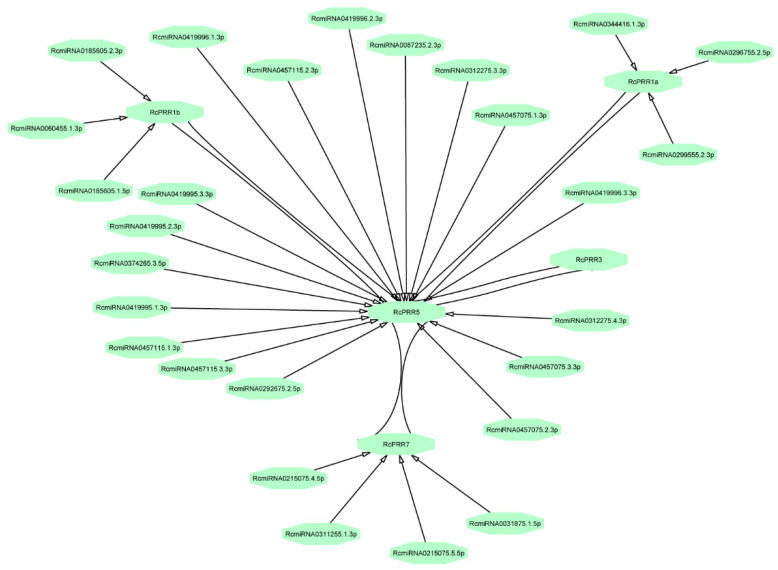
The interaction network of known *miRNA* of *Rosa chinensis* with clock *PRRs*. The network was performed by psRNAtarget tool and Cytoscape.

**Figure 6 ijms-23-07335-f006:**
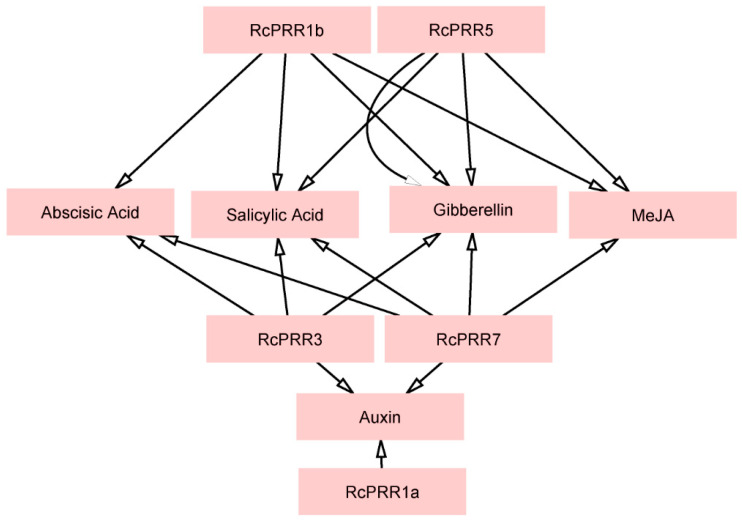
The interaction network of chemical compounds with clock *PRRs*. The network was performed by STRING and Cytoscape.

**Figure 7 ijms-23-07335-f007:**
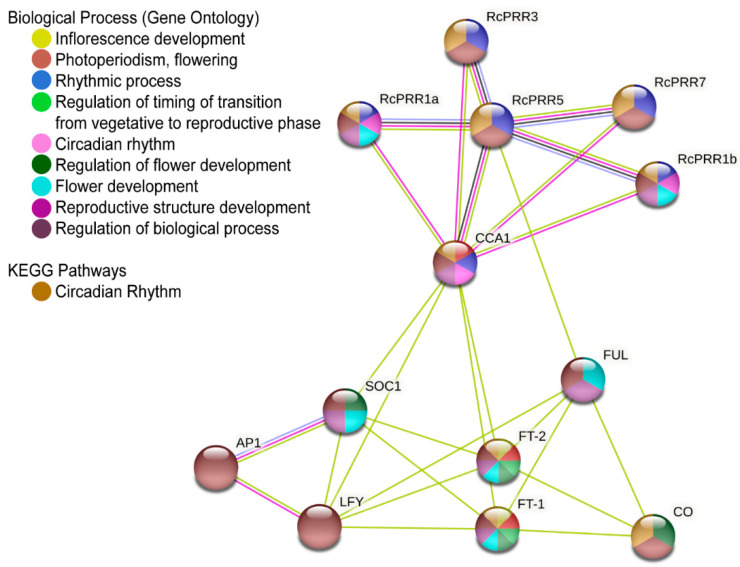
Co-expression network and functional annotation of clock *PRRs* genes with other flowering pathways genes in *Rosa chinensis.* Different colors of nodes represent genes sharing biological processes, while the edges/lines connecting the nodes represent the protein-to-protein interaction between genes. The addition of edges/lines between two nodes signifies a more significant interaction.

**Figure 8 ijms-23-07335-f008:**
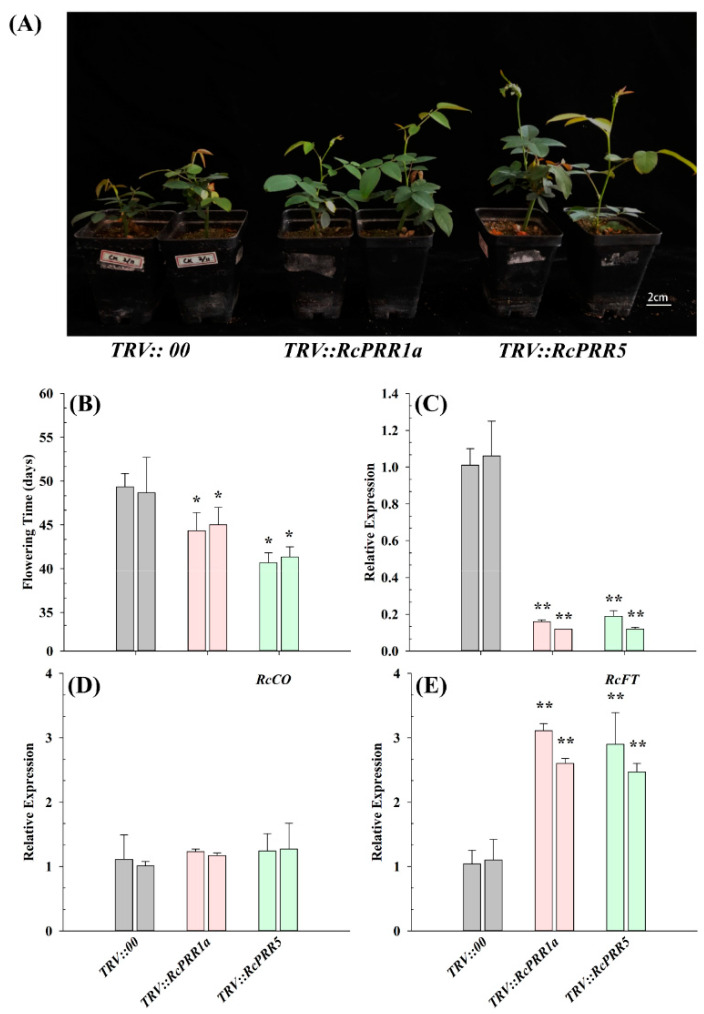
Silencing of *RcPRR1a* and *RcPRR5* in *Rosa chinensis*. (**A**) Flowering phenotype; (**B**) flowering time (days) of *Rosa chinensis*; and (**C**) expression levels of *RcPRR1a* and *RcPRR5* in two independent gene silenced lines; (**D**) expression levels of *RcCO* in two independent gene silenced lines; (**E**) expression levels of *RcFT* in two independent gene silenced lines. Gray bars, pink bars, and light green bars indicated the two independent lines of control, silenced lines of *RcPRR1a* and silenced lines of *RcPRR5,* respectively. Rose *GAPDH* gene was used as a reference. Three biological replicates were performed for each experiment. Asterisks above the bars indicate significant differences between gene silenced lines and the control as determined by the LSD test, ** *p* < 0.01 and * *p* < 0.05.

**Figure 9 ijms-23-07335-f009:**
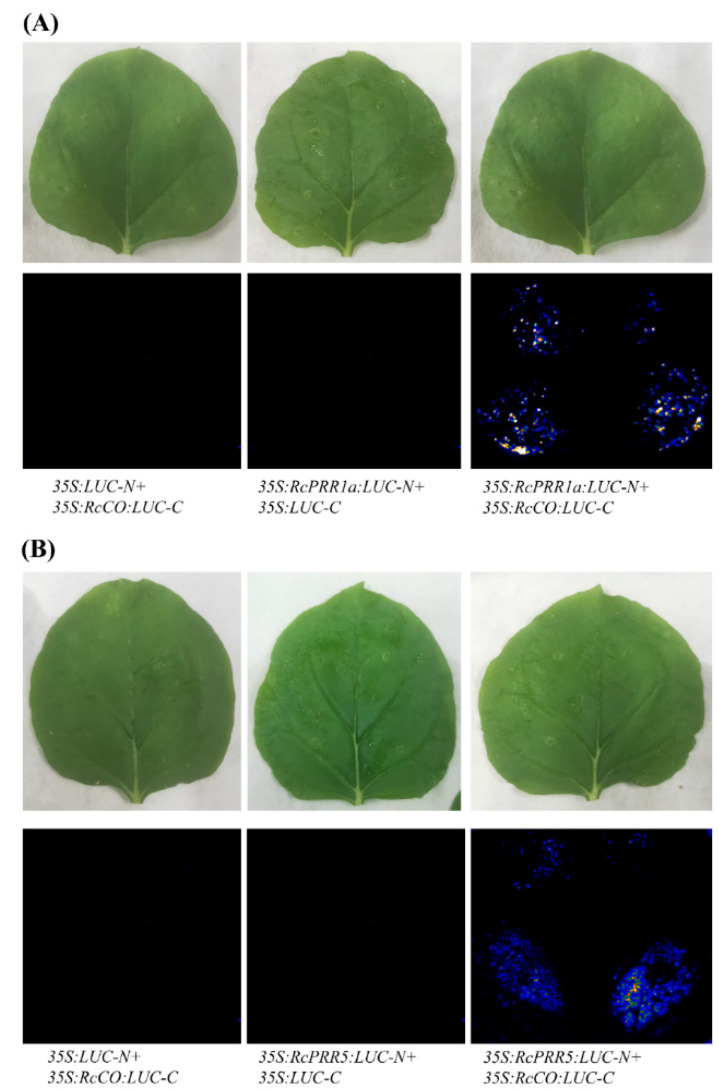
The protein-to-protein interactions of *RcPRR1a* and *RcPRR5* with *RcCO* in *Nicotiana benthamiana* leaves. (**A**) *35:LUC-N+35:RcCO:LUC-C* and *35:RcPRR1a:LUC-N+35:RcCO:LUC-C* represent the interaction of *LUC* with *RcCO* and *RcPRR1a,* respectively, while *35:RcPRR1a:LUC-N+35:RcCO:LUC-C* represent the interaction *RcPRR1a* with *RcCO* (**B**) *35:LUC-N+35:RcCO:LUC-C* and *35:RcPRR5:LUC-N+35:RcCO:LUC-C* represent the interaction of *LUC* with *RcCO* and *RcPRR5*, respectively, while *35:RcPRR5:LUC-N+35:RcCO:LUC-C* represent the interaction *RcPRR5* with *RcCO*.

**Table 1 ijms-23-07335-t001:** Non-synonymous (Ka) and synonymous (Ks) substitution rate and divergence time of clock *PRRs*.

Paralogous Genes		*Ka*	*Ks*	*Ka/Ks*	T (MYA)
*RcPRR1a*	*RcPRR5*	0.3587	0.558	0.642832	42.5
*RcPRR3*	*RcPRR7*	0.17805	0.33025	0.539137	25.1

## Data Availability

The data supporting the results are already mentioned in the main text and in Appendix A.

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
