# Peer review of "Evolutionary Analysis and Functional Identification of Clock-Associated PSEUDO-RESPONSE REGULATOR (PRRs) Genes in the Flowering Regulation of Roses"

_ijms, 2022, doi:10.3390/ijms23137335_

Round 1
Reviewer 1 Report
This article evaluated Evolutionary analysis and functional identification of clock associated PSEUDO RESPONSE REGULATOR (PRRs) genes in the flowering regulation of roses. This study will help to facilitate flower regulation in roses in future. Before recommending this article for publication, there are some shortcomings for that should be resolve.
General comments
Overall, the study is well designed and presented in a good way, but mostly the literature is not cited.
Abstract
Add methods in this section briefly.
Introduction
The introduction part is well written but still some details are required.
Add mechanism of circadian clock in rose plants.
Provide economic and commercial importance of the roses.
Why the authors selected rose to study PRRs
Results
Line 108-110 sentence is not clear must be revised.
Discussion
The authors correlated their results very well with previous studies. However, justification for each analysis should also be included.
Materials and Methods
Methodology is well presented.
Section 4.7 lack citation. The section can be cited with following article.
https://doi.org/10.3390/ijms22179175,
Conclusion
Conclusion is well justified but must be elaborated.
Author Response
1 Overall, the study is well designed and presented in a good way, but mostly the literature is not cited.
The paper is revised and the citations were added where missing.
2 Abstract-Add methods in this section briefly.
The methods are added in abstract section.
3 Introduction-The introduction part is well written but still some details are required.
Add mechanism of circadian clock in rose plants.
Provide economic and commercial importance of the roses.
Why the authors selected rose to study PRRs The missing details are added to the introduction part.
The Mechanism of the circadian clock in rose plants is still unknown, however the authors added the mechanism of circadian clock in Arabidopsis to support the current study.
Economic and commercial importance of roses are added.
PRRs are not yet studied in roses, and the purpose of selection rose for studying PRRs is mentioned in introduction section.
4 Results-Line 108-110 sentence is not clear must be revised.
The sentence has been revised.
5 Discussion-The authors correlated their results very well with previous studies. However, justification for each analysis should also be included.
Justification for each analysis has been added (where missing) in the discussion section.
6 Materials and Methods-Methodology is well presented.
Section 4.7 lack citation. The section can be cited with following article.
https://doi.org/10.3390/ijms22179175, The reference is added to the prescribed section, all the references in materials and methods section are revised.
The given reference is cited in the revised version.
7 Conclusion-Conclusion is well justified but must be elaborated.
The conclusion is elaborated precisely.
Reviewer 2 Report
The article entitled, “Evolutionary analysis and functional identification of clock associated PSEUDO RESPONSE REGULATOR (PRRs) genes in the flowering regulation of roses” is well written and concludes role of selected PRR in flowering in Rosa chinensis. This article can be accepted only after major revision, correcting the some grammatical/typographical errors along with better resolution and good representative pictures.
Suggested minor revision-
- There are some grammatical errors in the manuscript which need to be corrected after thorough reading.
- Few of the lines mentioned for examples-
· Line no. 9-10: Rephrasing is required. For example, “genome wide identification of clock PRRs and their….”
· Line no. 53: comma after “In this study”.
There are some suggestions-
· What was the criterion of choosing basal angiosperms, rosids and rosaceae for multiple sequence alignment? You could have chosen closely related species for the evolutionary analysis. Further, the updated version of MEGA software can be used.
· The domain analysis should be carried out using multiple software for better result. In the present study only single software has been used. Therefore, it is suggested to analyze the domain architecture using other software’s like SMART etc.
· What were the parameters used for the motif analysis.
· The authors are advised to include the interaction of the genes with the chemical compounds using STRING database.
Ø Also, the representative picture of VIGS is not clear, please provide better picture or close up pictures for phenotypes if available.
Ø Figure 5 is not clearly mentioning the names of the genes and control. It is confusing and should be clearly mentioned. Also, the figure legend should be explained properly. The image 5A resolution can be improved.
Ø The legend of figure 6 should be clearly explained, it is not understandable.
Author Response
1 Line no. 9-10: Rephrasing is required. For example, “genome wide identification of clock PRRs and their….”
The sentence has been revised.
2 Line no. 53: comma after “In this study”.
The sentence has been revised.
3 What was the criterion of choosing basal angiosperms, Rosids and Rosaceae for multiple sequence alignment? You could have chosen closely related species for the evolutionary analysis.
Further, the updated version of MEGA software can be used. Related species of Rosaceae were used for evolutionary analysis. Rosids is a large group of angiosperms containing Vitals (Vitaceae), Fabids (Rosaceae), and Malvids (Brasicaceae). The evolutionary analysis were carried out for Basal angiosperms vs Rosids. Within the Rosids, the species were selected from each sub group to get the profound results for Rosaceae.
It has been revised in materials and methods section to remove the confusion.
The updated version of MEGA software (MEGA 11) is used to reconstruct the phylogenetic tree for roses.
4 The domain analysis should be carried out using multiple software for better result.
In the present study only single software has been used. Therefore, it is suggested to analyze the domain architecture using other software’s like SMART etc. Domain analysis were carried out using multiple software (i.e. Pfam, SMART, NCBI CDD, and MOTIF search) and is mentioned in the materials and methods section for the identification of PRR orthologous. The confirmed sequences were then used for structural organization of domains in NCBI CDD and visualized via TBtools.
5 What were the parameters used for the motif analysis.
The default parameters of MEME suit software were used for motif analysis and has been mentioned in the materials and methods section.
6 The authors are advised to include the interaction of the genes with the chemical compounds using STRING database.
String database could not provide the interaction of genes with chemical compounds rather it only provide to protein to protein interaction and the shared biological and molecular function that has been done by authors. However, Stitch database (http://stitch.embl.de/) could provide the interaction of genes with chemicals. The authors searched stitch database for the possible interaction of PRR genes with chemical compounds but unfortunately there was no data available for Rose plants (as these genes has never been studied previously in roses). Moreover, the interaction of chemical compounds with genes are often used in the studies on pathogens/diseases that is beyond our focus.
7 Also, the representative picture of VIGS is not clear, please provide better picture or close up pictures for phenotypes if available.
The close up pictures for phenotypes is added to the figure.
8 Figure 5 is not clearly mentioning the names of the genes and control. It is confusing and should be clearly mentioned. Also, the figure legend should be explained properly. The image 5A resolution can be improved.
The resolution of the figure has been improved, the legend is updated and name of the gene and control is clearly mentioned.
9 The legend of figure 6 should be clearly explained, it is not understandable.
The legend of Figure 6 has been revised and clearly explained.
Round 2
Reviewer 1 Report
First, I would like to congratulate the authors, because the manuscript, is well revised and has an important objective to be achieved.
Author Response
All the authors are very grateful to the reviewer for his/her valuable comments. It helped us very much to improve our paper.
Reviewer 2 Report
4.2. Clock PRRs Orthologous Identification in Roses and Angiosperms
The Arabidopsis clock associated PRRs protein sequences i.e. PRR1 (At5G61380), 295 PRR3 (At5G60100), PRR5 (At5G24470), PRR7 (At5G02810), and PRR9 (At2G46790), were used as queries to search against the whole genomes of all the.......
???? Is BLAST search done against the whole genome? or Protein model sequences? Which Blast has been used, should be clarified.
Why only arabidopsis sequences are used? Authors may get additional genes by using the PRR sequences of other plant sps also.
Is E-value threshold of 1e-3 significant? Usually people use 1e-10 for these analysis.
Why authors have not used HMM profile search, which is the best method for identification of new proteins in non-model plants.
4.5. Co-expression network organization.....
STRING doesn't provide a co-expression network, It should be corrected accordingly.
Further, the Cytoscape extension of STRING can be used for identifying the interacting chemical compounds besides the Stitch which is very limited.
Since these are sensor proteins, they must interact with certain signal molecules.
Further, authors should also analyze the Physico-chemical characteristics of the identified PRR proteins and genes, along with subcellular localization, at least by using the in-silico methods.
The title of Ms starts with Evolutionary analysis, while I could not see any such analysis except phylogeny. They should also perform additional evolutionary analyses including Ka/Ks, Tajima test etc. They may follow and cite these Ms for the above analyses. https://www.mdpi.com/2223-7747/11/5/587 ; https://www.mdpi.com/2223-7747/11/7/911
Interaction analysis of known miRNAs can also be included to get insight into the mechanism.
Figure 3 legends: Are these domains and motifs discovered by authors? or identified in the PRR sequences?
All these kind of sentences needs to be corrected. and What is the significance of showing weblogo of motifs? Are these motif belong to any functional or important domain? if yes, then it should be included in Ms.
The discussion part is not up to the mark, It lacks several parts. Authors should discuss each of their findings here.
Abstract and conclusion should be rewritten after the suggested revision of the Ms.
There are numerous loopholes in the experimentation and Ms writing. I can not recommend its publication in an esteemed journal like IJMS at this stage. It must undergo major revision before publication.
Author Response
1 4.2. Clock PRRs Orthologous Identification in Roses and Angiosperms
The Arabidopsis clock associated PRRs protein sequences i.e. PRR1 (At5G61380), 295 PRR3 (At5G60100), PRR5 (At5G24470), PRR7 (At5G02810), and PRR9 (At2G46790), were used as queries to search against the whole genomes of all the.......
???? Is BLAST search done against the whole genome? or Protein model sequences? Which Blast has been used, should be clarified.
Why only arabidopsis sequences are used? Authors may get additional genes by using the PRR sequences of other plant sps also.
Is E-value threshold of 1e-3 significant? Usually people use 1e-10 for these analysis.
Why authors have not used HMM profile search, which is the best method for identification of new proteins in non-model plants. The authors have used the method for the identification of clock PRRs as previously by Han et al., 2019.
The sentences for doing BLAST-P and using protein sequences set are revised and the confusion has been removed.
As the Arabidopsis sequences were used as a queries to find all the PRRs in other species, a reciprocal BLAST were also performed for each specie separately to confirm the valid results for the identified sequences.
The authors confirmed the significance if E-value 1e-3 and it has previously used by Han et al., 2019. As the authors were following the methods of Han et al., 2019, so we have taken the E-value 1e-3.
2 4.5. Co-expression network organization.....
STRING doesn't provide a co-expression network, It should be corrected accordingly.
Further, the Cytoscape extension of STRING can be used for identifying the interacting chemical compounds besides the Stitch which is very limited.
Since these are sensor proteins, they must interact with certain signal molecules.
Interaction analysis of known miRNAs can also be included to get insight into the mechanism.
Further, authors should also analyze the Physico-chemical characteristics of the identified PRR proteins and genes, along with subcellular localization, at least by using the in-silico methods.
The title of Ms starts with Evolutionary analysis, while I could not see any such analysis except phylogeny. They should also perform additional evolutionary analyses including Ka/Ks, Tajima test etc. They may follow and cite these Ms for the above analyses. https://www.mdpi.com/2223-7747/11/5/587 ; https://www.mdpi.com/2223-7747/11/7/911
The authors are very grateful to the reviewer for the valuable suggestion and sheer direction.
Cytoscape extension of STRING have used for making the interaction network of PRRs with chemical compounds and miRNAs.
A new section (2.5) is added for the interaction of PRRs with chemical compounds and known miRNAs.
The authors have already analysed the Physico-chemical characteristics of the identified PRR proteins and genes and subcellular localization. It is mentioned in section 2.3 of manuscript.
The given referenced helped the authors a lot to do the further analysis. The authors cited these papers and the
Evolutionary analyses like Ka/Ks and Tajima test were performed along with the cis-regulatory elements and explained in the newly added sections (2.4) of the manuscript.
3 Figure 3 legends: Are these domains and motifs discovered by authors? or identified in the PRR sequences?
All these kind of sentences needs to be corrected. and What is the significance of showing web logo of motifs? Are these motif belong to any functional or important domain? if yes, then it should be included in Ms.
The sentences has been revised carefully and the confusion has been removed.
The explanation for motif were added to the results (section 2.3).
4 The discussion part is not up to the mark, It lacks several parts. Authors should discuss each of their findings here.
The discussion has been revised accordingly.
5 Abstract and conclusion should be rewritten after the suggested revision of the Ms.
The abstract and conclusion have been revised accordingly.
6 There are numerous loopholes in the experimentation and Ms writing. I cannot recommend its publication in an esteemed journal like IJMS at this stage. It must undergo major revision before publication.
After the extensive editing and addition of certain analysis according to the reviewers’ comments, the authors feel confident that the revised manuscript will satisfy the requirements of publication.
Round 3
Reviewer 2 Report
Authors have addressed all the concerns.